# Early Warning Early Action for the Banking Solvency Risk in the COVID-19 Pandemic Era: A Case Study of Indonesia

**Taufiq Hidayat [1,]*** [ID], **Dian Masyita [2], Sulaeman Rahman Nidar [2], Fauzan Ahmad [3] and Muhammad Adrissa Nur Syarif [3]**

1 STIE Indonesia Banking School, Jakarta 12730, Indonesia
2 Fakultas Ekonomi dan Bisnis, Universitas Padjadjaran, Bandung 40132, Indonesia; dian.masyita@unpad.ac.id (D.M.); sulaeman.rahman@unpad.ac.id (S.R.N.)
3 System Dynamics Bandung Bootcamp, Bandung 40534, Indonesia; ojan_154@yahoo.com (F.A.); adrisansyarif@gmail.com (M.A.N.S.)
* Correspondence: taufiq.hidayat@ibs.ac.id

**Abstract:** The COVID-19 pandemic has affected people's lives and increased the banking solvency risk. This research aimed to build an early warning and early action simulation model to mitigate the solvency risk using the system dynamics methodology and the Powersim Studio 10© software. The addition of an early action simulation updates the existing early warning model. Through this model, the effect of policy design and options on potential solvency risks is known before implementation. The trials conducted at Bank BRI (BBRI) and Bank Mandiri (BMRI) showed that the model had the ability to provide an early warning of the potential increase in bank solvency risk when the loan restructuring policy is revoked. It also simulates the effectiveness of management's policy options to mitigate these risks. This research used publicly accessible banking data and analysis. Bank management could also take advantage of this model through a self-stimulation facility developed in this study to accommodate their needs using the internal data.

**Keywords:** banking sector; COVID-19 outbreak; corporate insolvency; simulation; loan restructuring policy; system dynamics; early warning early action

## 1. Introduction

The COVID-19 pandemic has impacted all sectors and the economic activities of people in Indonesia. Almost all economic activities experienced a slowdown, especially businesses supported by loans from banks, including large enterprises and small and medium enterprises (SMEs). Consequently, these conditions have impacted the banking sector, resulting in several risks. These include an increased risk of non-performing loans (Barua and Barua 2021), abundant bank liquidity due to slowing loan demand (Goodell 2020), and a decline in banking profitability (Knowles et al. 2020). Although the economic crisis due to the COVID-19 outbreak is different from previous crises, it has similar consequences, including widespread business bankruptcies, increased unemployment, and worsening banking solvency (Danielsson et al. 2020).

The Financial Services Authority (OJK), as the supervisory authority and regulator of banking in Indonesia, issued a loan restructuring policy for debtors that were affected by the pandemic in March 2020 to reduce the pandemic's impact on solvency risk. The non-performing loan (NPL) restructured based on this regulation can still be categorized as a performing loan, and the bank does not have to set aside any loan impairment expenses. The policy is a quick response to the impact of COVID-19, relaxing the rules for restructuring non-performing loans, and was enforced for the first time. It was extended several times, until 31 March 2023.

The restructuring policy aims to curb the increase in NPL and allow time for banks to strengthen their reserves for impairment losses on loans and capital to avoid the solvency

risk. However, the NPL ratio of banks in Indonesia reached 3.35% in June 2021, the highest level since January 2019. This could still increase if outstanding loans that were restructured with the OJK policy are not entirely repaid. As of 30 June 2021, the proportion of restructured loans to total loans was 17.32% (Bank Indonesia 2021). Siregar et al. (2021) estimated that the NPL potential of the loan restructuring would range from 10 to 30% when the OJK policy is revoked. In these estimations, the national banking industry experiences a potential NPL ratio level above 5% or exceeds the NPL standard set by the regulator. Subsequently, the capital adequacy ratio (CAR) would decline, increasing the bank solvency risk.

In reference to this banking situation, the challenge is to estimate the bank solvency conditions based on each bank's capital capacity and potential NPL risk. Considering the outstanding loan restructures, NPL volume and ratio, the simulation for each bank could be calculated to detect the risk of solvency at an early stage (early warning), and present some early action plans (early action) to prevent the risk (Leaning 2016; Lang et al. 2018). Al-Kharusi and Murthy (2020) and Pavlov and Katsamakas (2021) examined early warning financial risks during the COVID-19 pandemic using a financial statement simulation methodology. However, the research did not model an early action plan to prevent these potential risks. Therefore, this research developed the model's function to not only provide some early warnings but also propose some early actions in one simulation, to prevent the bank solvency risk related to regulation.

The system dynamics methodology was used to model the complex financial transactions to produce financial statement baselines. Additionally, the methodology helps to simulate the feedback effect of the early actions with some changes to the baseline financial statement behavior (Oladimeji et al. 2020). Subsequently, this research produces an early information simulation model of solvency risk per individual bank and preventive policy options for management of the potential risks. The model could practically become a self-simulation tool for early warning and early action regarding solvency risks in bank management.

The research questions of this study are: (1) How to develop the simulation model to provide an early warning of solvency risk in the banking industry in Indonesia during the COVID-19 outbreak; (2) How to simulate early action to mitigate and prevent the solvency risk to response the revocation of the loan restructuring policy by OJK. Using system dynamic simulation, this research resulted in a model that can deliver early warnings for individual bank solvency risk and early actions for the individual bank management to prevent the solvency risks and increase the readiness for revocation of the loan restructuring policy.

This research makes an important contribution to the study of the early warning system of bank bankruptcy risks for several reasons, namely: (1) the use of system dynamics simulation methods to modelling the complex, dynamic and ongoing bank risk behaviour during the pandemic COVID-19, (2) the research is to be able to produce an effective early warning of the bank solvency risks and (3) the research is a kind of a forward looking oriented simulation to predict the potential bank bankruptcy risks. The added value of this research is the existence of a dynamic bank balance sheet simulation so that the condition of the bank's asset, liabilities, capital and profit and loss during the COVID-19 pandemic and the risk of bank solvency can be detected at any time. By monitoring the condition of bank's loan performance through several main variables related to solvency risk, bank's management can determine appropriate policies to reduce this risk and indirectly reduce the risk of bank's bankruptcy. This research is able to produce simulation models in the form of early warnings and simulations of several policy options as early actions for the bank's management to response the risk of non-performing loan, solvency risk and bank's bankruptcy due to the COVID-19 pandemic.

The structure of this paper consists of five parts, beginning with the introduction that contains the background of the research. Section 2 discusses the literature review, namely the related literature and previous research, Section 3 explains the research methodology followed by the research results and Section 4 is a discussion of the results of the research.

Lastly is the section of the conclusion and suggestions that also contains the implication of the research results. This study is ended with a reference of the studies that used in the research.

## 2. Literature Review

According to De Vany (1984), the main causes of bank bankruptcy are the information asymmetry, agency problem and moral hazard that occur together. Smith (2010) found evidence that there is a correlation between agency problems and bank bankruptcy in the crisis period of 2007 and 2008. The agency problem is the problem of mismatch of interests between shareholders as principals and management as agents (Jensen and Meckling 1976; Rose 1992). In the banking sector, agency relationships occur between bank management with shareholders and banking supervisory authorities (Henrard and Olieslagers 2004) as well as with depositors (Kuritzkes et al. 2003). Banking supervisory authorities play a role in protecting the interests of depositors by issuing various regulations that must be obeyed by bank management and shareholders (Donnellan and Rutledge 2016), including the issuance of policies for loan restructuring during the COVID-19 pandemic by the Financial Services Authority, then it is an agency relationship intervention in order to reduce the risk of bank bankruptcy and protect depositors (Hidayat et al. 2021).

Bank bankruptcy can also arise due to changes in financial conditions both internally and externally of the bank. Bank bankruptcy can also arise from increased loan risk arising from debtor moral hazard, weak analysis of creditworthiness, external conditions such as the decline in the community's economic capacity due to the COVID-19 pandemic or lending to high-risk sectors. Bank management as an agent in the agency theory needs to recognize weak signals in the economic environment that will affect loan risk and bank bankruptcy, such as high NPL levels and declining CAR ratios. A weak signal is a symptom of bank performance that provides the basis for managerial decision making to ensure that the bank's strategy can be achieved. Meanwhile, based on the weak banking signal information, a system is needed to provide some early warnings for bank management to be aware of the potential risks that may arise. early warning systems are a key tool for bank management to anticipate and make policies to reduce the potential risks of bank bankruptcy (Gunnersen 2014).

Most of the literature research on early warnings of bank solvency risk are backward-oriented. This means that the literature is based on historical financial statements to provide early warning indicators of these potential risks. For instance, Korzeb and Niedziółka (2020); Barua and Barua (2021); Hardiyanti and Aziz (2021) showed the phenomenon of increased NPL risk during the COVID-19 pandemic. The increase in NPL reduced cash flow, profit, and CAR (Mayes and Stremmel 2012; Donnellan and Rutledge 2016). Consequently, a decrease in CAR increased the bank solvency risk, as measured by Z-Score (Lepetit et al. 2020). This research used the NPL, CAR, and Z-Score ratios to show the solvency risk.

Facing the COVID-19 pandemic requires a forward-looking approach for early warnings of potential bank solvency risks and early action to prevent these risks. For these reasons, the simulation methodology was used to develop baseline projections for financial statements. These baselines describe future financial risk conditions and changes in their behavior due to some unexpected events. Furthermore, the simulation results were used to develop several alternative early action policies to reduce solvency risk (Pavlov and Katsamakas 2021; Petropoulos et al. 2020). These include the promotion of loan growth, managing restructured loans, increasing bank operational cost efficiency, lowering interest expenses, and increasing loan interest (Bastana et al. 2016; Samorodov et al. 2019; Rahmi and Sumirat 2021). Those policies are expected to strengthen bank capital to avoid solvency risk. However, the early action policy should be simulated first to determine the feedback on potential changes in the solvency risk levels and the optimum policy options (Schuermann 2014). According to Wu (2014) and Kunc et al. (2018), the system dynamics methodology is a simulation modeling that accommodates the feedback process in managerial decision making.

System Dynamics is a methodology to design strategies and policies with computer simulation tools (Sapiri et al. 2020), to produce better responses to the complex and dynamic problems in the social, managerial or economic fields (Sterman 2000; Morecroft 2015; Duggan 2016). To solve the complex problems, according to Bala et al. (2017), the structures, and the relationship between the structures in the problem, should be analyzed. The system dynamics model describes the structure of financial statement accounts based on stock, rate, auxiliary, and constant. The pattern of the relationships between the accounts in the financial statements are modeled through causal-loop and stock-flow diagrams (García 2019). The financial statements model of system dynamics was developed in some studies by Islam et al. (2013); Wu (2014); Istiaq (2015); Pierson (2020); Aksu and Tursun (2021); Pavlov and Katsamakas (2021); Hidayat et al. (2021) to analyze financial reporting, banking risk management, management control systems, and solvency stress-testing.

## 3. Methodology

This research used the previous literature to accommodate the bank's needs when dealing with the COVID-19 pandemic and the loan restructuring policy revocation. It aims to develop a simulation model to generate early warnings and identify early actions to mitigate bank solvency risks using Powersim Professional 10© software. Furthermore, it intends to advance the modeling of bank financial statements and several important financial ratios developed by Islam et al. (2013); Pierson (2020) and Hidayat et al. (2021) through causal-loop and stock flow diagrams.

The modeling begins by analyzing the structures of bank financial statements and growth assumptions to produce the baseline financial report, and then could be used as a comparison in further simulations. When the loan restructuring policy ends, the baseline financial report is simulated to obtain the early warning on solvency risk and prepare the policy response. The policy response is simulated first to determine its impact on solvency risk before implementation. The simulation modeling stages for early warning and early action for bank solvency risk are presented in Figure 1.

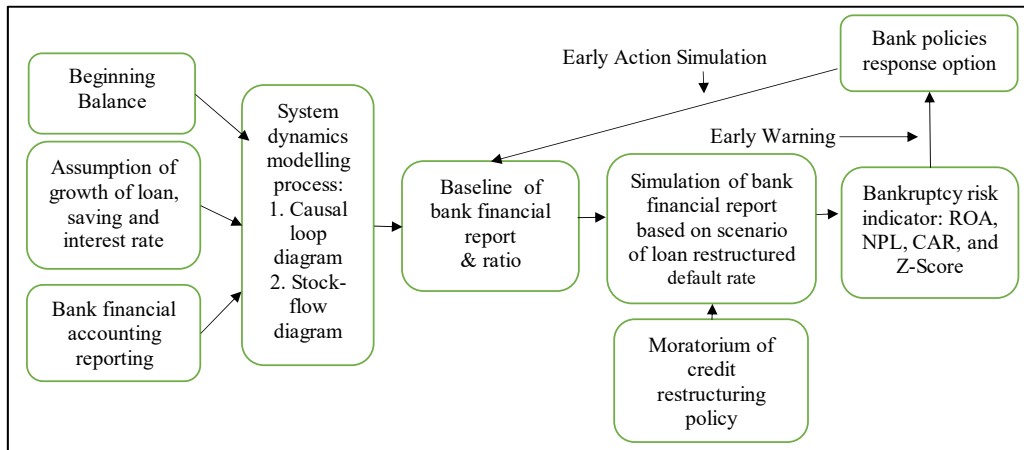

**Figure 1.** Stages of Early Warning, Early Action Simulation Modeling of Bank Solvency Risk (Schuermann 2014).

The simulation model was applied to BBRI and BMRI, the two largest banks in Indonesia. Data were obtained on the initial balance of financial statements to model from the financial position statement balance as of 31 December 2019 (audited). The simulation modeling period is monthly, from the 1st period (January 2020) to the 48th period (December 2023). An early warning simulation of solvency risk was performed in the 21st period, or September 2021, to determine the early risk condition if the loan restructuring policy revocation is implemented in March 2023 (39th period). After obtaining the early warning information, an early action simulation was performed to determine the

impact of each policy on changes in CAR and Z-Score until the 48th period. The bank is categorized as insolvable if the Z-Score < 0 and/or the CAR ratio less than its threshold based on the standard of each bank. The higher the CAR and Z-Score, the more solvable the bank (Lepetit et al. 2020). The Z-Score formula used was:

$$\text{Z-Score: } ((\text{ROA} + (\text{equity}/\text{total assets}))/\text{ROA standard deviation} \qquad (1)$$

The other financial ratios that become indicators of bank solvency are:

$$\text{Return on asset (ROA)} = \frac{\text{Net Profit}}{\text{Total Asset}} \qquad (2)$$

$$\text{Capital adequacy ratio (CAR)} = \frac{\text{Equity}}{\text{Total risk-weighted asset}} \qquad (3)$$

$$\text{Non-performing loan (NPL) ratio} = \frac{\text{Non-performing}}{\text{total loan}} \qquad (4)$$

$$\text{Loan loss provision (LLP) ratio} = \frac{\text{Loan loss provision}}{\text{Non-performing loan}} \qquad (5)$$

### 3.1. Causal Loop Diagram of Early Warning Early Action Simulation Model

The structure of the simulation model was prepared based on the account components in the financial statements, transaction flows to earn profits, early information on bank solvency risks, and early actions to mitigate these risks. This was described as a causal-loop diagram in Figure 2 to show the transaction flow of the banking business activities. The positive link shows a unidirectional or positive causality relationship between the two structures. In contrast, a negative link shows an inverse or negative causality relationship between the two structures.

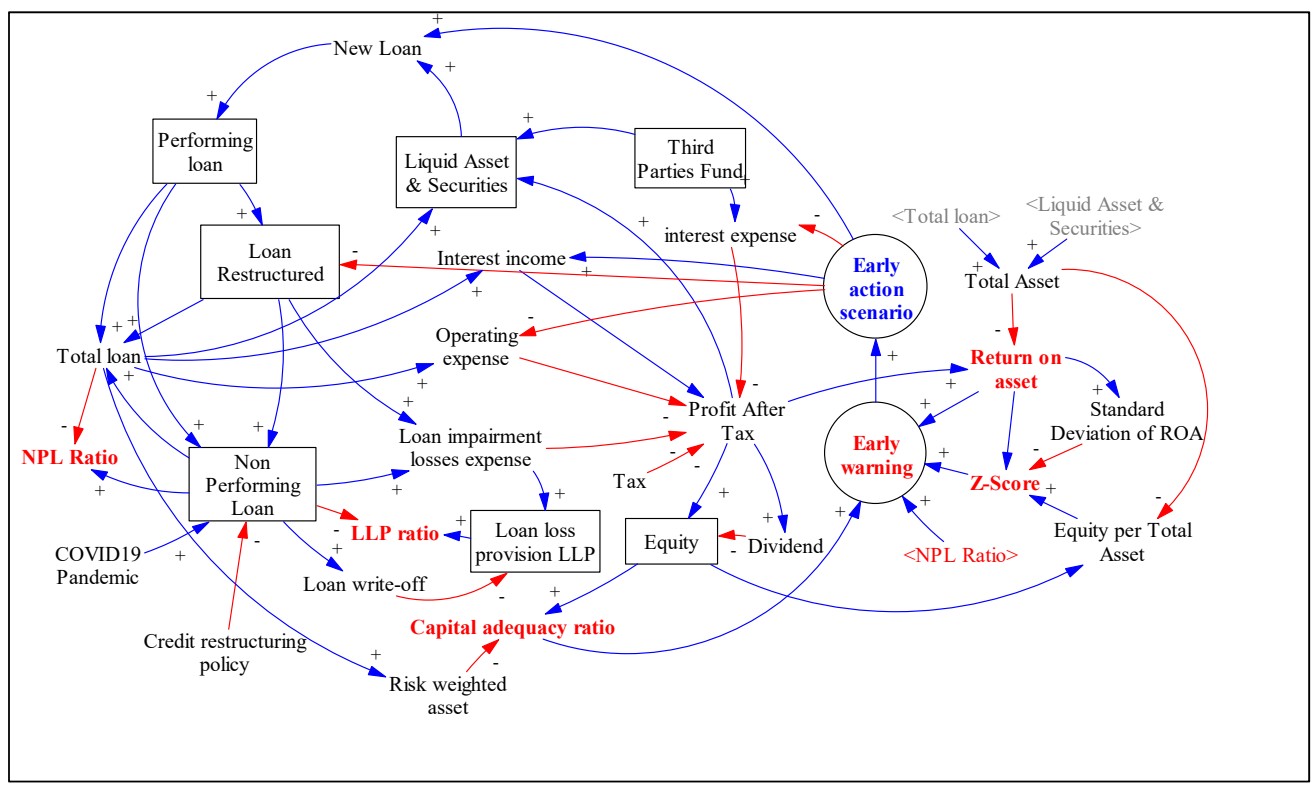

**Figure 2.** Causal-loop Diagram of Early Warning Early Action Model (Source: Author). The + sign indicates a unidirectional relationship between two variables, while the − sign indicates an inverse relationship between the two variables

When conducting the intermediary function, the banking business receives third-party funds (TPF) or savings that could be converted into loans to obtain an interest income. When interest income covers the interest expense, operating expense, loan impairment losses expenses and tax, the bank reports some net income as additional capital. During the COVID-19 pandemic crisis, interest income decreased, while loan impairment losses increased, meaning that banks faced an increased solvency risk.

The early warning indicators of banking solvency risk used in this model appear in terms of ROA, NPL Ratio, CAR, and Z-Score. Any weakening in these indicators forms the basis for determining early action policy scenarios to improve the conditions. Therefore, this research simulated early action scenarios to overcome the potential weakening of banking solvency risks when the loan restructuring policy is revoked in March 2023. The scenarios include:

a. Promoting loan growth or new loan policy to increase performing loans, to obtain more interest income and strengthen ROA, CAR, and Z-Score. However, when economic growth is abnormal, new loans should be selectively added to avoid additional NPL.

b. Interest management is carried out by adjusting the loan interest rate and the savings interest rate to obtain an optimum net interest margin.

c. The efficiency of operating expenses, including bank overhead, employee costs, and other expenses could reduce the ratio of operating expenses to income.

d. Combined policy of (a), (b) and (c) above.

### 3.2. Stock Flow Diagram of Early Warning, Early Action Simulation Model

For simulation purposes, the relationship between the structure of financial transactions depicted in the causal-loop diagram in Figure 2 was operationalized into a stock-flow model, whose symbol is presented in Table 1. The structure of the balance sheet in the financial statements, consisting of equity, performing loans, third party funds and others in Figure 2, are categorized as stock because they have a balance at a certain time. In this situation, changes in stock balance are determined by the rate of inflows and outflows per unit time. The rate is determined by the multiplication of the stock and a constant variable, either directly or through several calculation stages using the auxiliary function.

**Table 1.** Symbols of Stock-Flow Diagram in Powersim Professional 10© (Sterman 2000).

| Symbol | Definition |
| --- | --- |
|  | The symbol of STOCK is to declare variables with an accumulation derived from the previous value plus the difference between inflows and outflows. $\text{Stock}(t) = \int_{t0}^{t1} = (\text{Inflow(s)} - \text{Outflow(s)})ds + \text{Stock}(t_0)$ $d(\text{Stock})/dt = \text{Inflow}(t) - \text{Outflow}(t)$. |
|  Rate of loan market | The symbol of RATE states the formulation of the amount of the stock inflow and outflow in the system in a certain time unit. For example, the rate of loan market = \$100/month. |
|  Constant 1  Aux B　Aux | The symbol of AUXILIARY or AUX is used to formulate the equation rate by defining the determining factors of the rate equation separately. Additional equations are substituted for each other and several separate rate equations. For example, Aux = Aux B x Constant |

**Table 1.** *Cont.*

| Symbol | Definition |
|---|---|
| Constant_1 | The symbol of CONSTANT is a function of a certain number, the input for the auxiliary or equation rate in the model, its value remains in the simulation period. It is used to simulate management policies, such as loan interest rate income and saving interest rate expense policies. |
| ⟶ | The symbol of ARROW indicates the flow of information from one variable (auxiliary, stock, constant, level) to another. |
| Target loan to deposit ratio LDR assumption | The symbol of GRAPH contains certain parameter functions to explain other parameters/quantities. |

Stock-flow diagrams are developed based on bank financial statement accounts, including the financial assets account group (Figure 3), liabilities account group (Figure 4), and equity and statement of profit or loss (Figure 5). The formulation used to calculate the changes in the account is given in Appendix A.

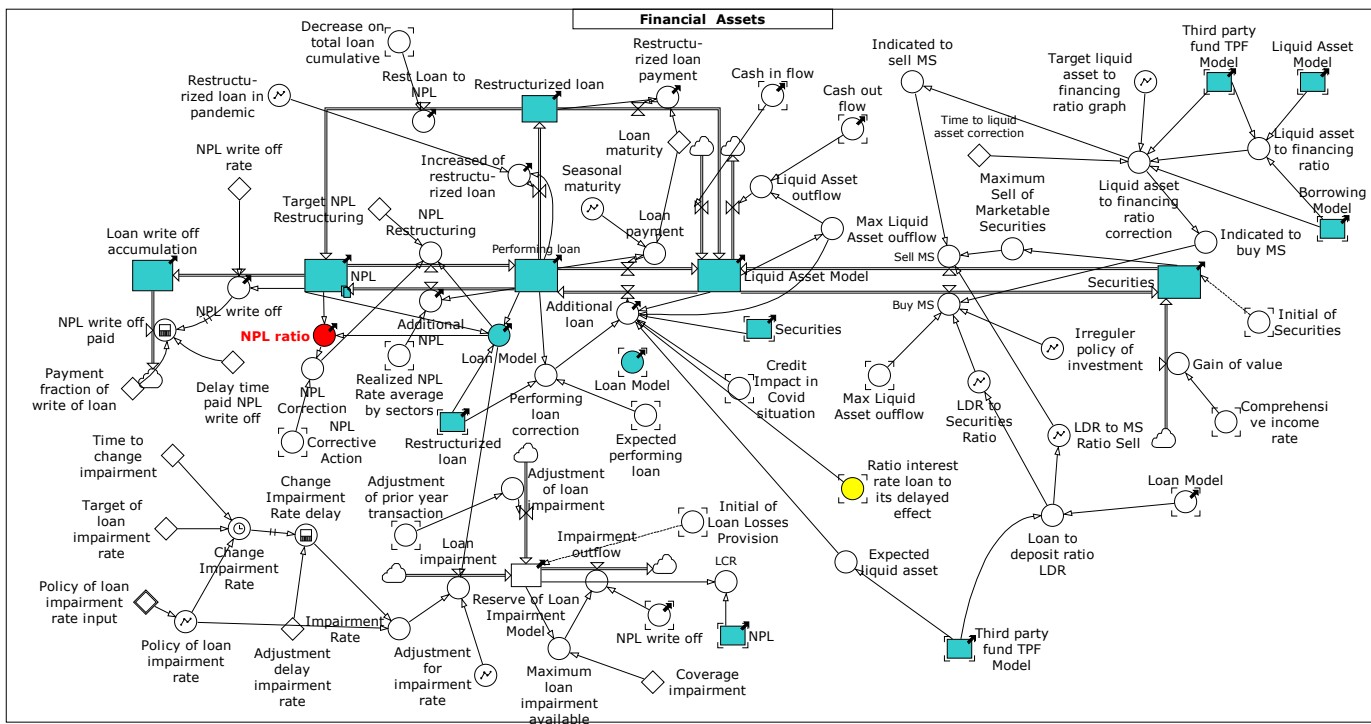

**Figure 3.** Stock-flow Diagram of Financial Assets (Source: Author).

The financial assets account diagram shows the cycle of investment transactions to earn interest income and maintain liquidity. In this situation, the bank prioritizes its liquid assets for investment in performing loans and generating high returns, although it is necessary to anticipate the risk of loan default. A defaulted loan then could be restructured and controlled as a restructured loan, while a defaulted loan that could not be restructured could then be administered as an NPL. Since eliminating NPL write-off reduces capital, the bank forms a loan lost provision (LLP). The NPL ratio is the early warning of loan risk, which must not exceed 5%. The bank maintains adequate levels of liquidity in the form of liquid assets and securities. The total balance between the two financial assets can meet the transaction payment needs for the next 1 month. Investment in securities generates interest income.

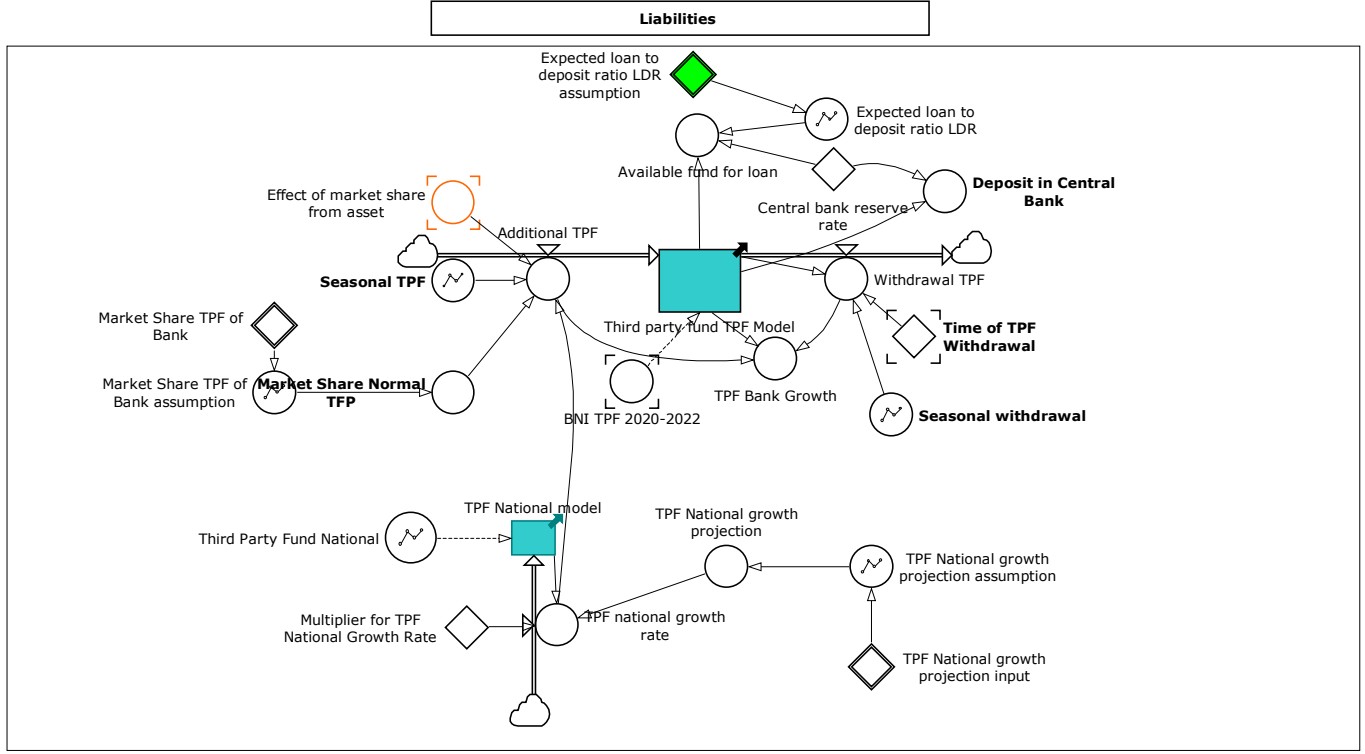

**Figure 4.** Stock-flow Diagram of Liabilities (Source: Author).

In the research design, the impact of changes in loan interest rates on additional loans is explained in the function of the interest rate loan to its delayed effect ratio variable in the additional loan formula in Table A1. If the annual interest rate of loan is increased, it will have a negative impact on additional loans.

The stock-flow diagram of liabilities illustrates the transaction flow of the third-party fund (TPF) originating from the addition and withdrawal of TPF funds from savers, as presented in Figure 4. An additional TPF is determined based on the national savings fund growth and the bank's market share. After deducting the allocation of statutory reserves requirements at Bank Indonesia, the remaining TPF funds became available funds for new loans to debtors.

Furthermore, for deposit growth, we use the variable size of bank assets. This is based on the Indonesia Banking Survey 2017 conducted by PWC. It can be seen that the amount of bank assets has a strong correlation with additional third-party funds. Furthermore, additional loans at Bank BRI and Bank Mandiri are encouraged because they have more extensive networks and access to customers. Therefore, in this study the effect of market share from assets is used as a reinforcing variable for additional third-party funds. Moreover, most of the third-party funds of Bank BRI and Bank Mandiri come from the government institutions and state-owned companies which are not sensitive to the amount of interest rates on deposit. The explanation is shown in the Figure 4.

The stock-flow diagram of profit, loss, and equity in Figure 5 is developed through the accounts that make up income and expenses, as well as other equity transactions. Income is calculated before and after tax, which regularly changes following the banking business model. However, there some irregularly transactions that change the equity, including buy-back of stock, adjustments of expenses for transactions in previous years, employment benefit adjustments, and gains or losses of financial assets. The indicators of bank solvency risk that were analyzed in this paper are CAR and Z-Score.

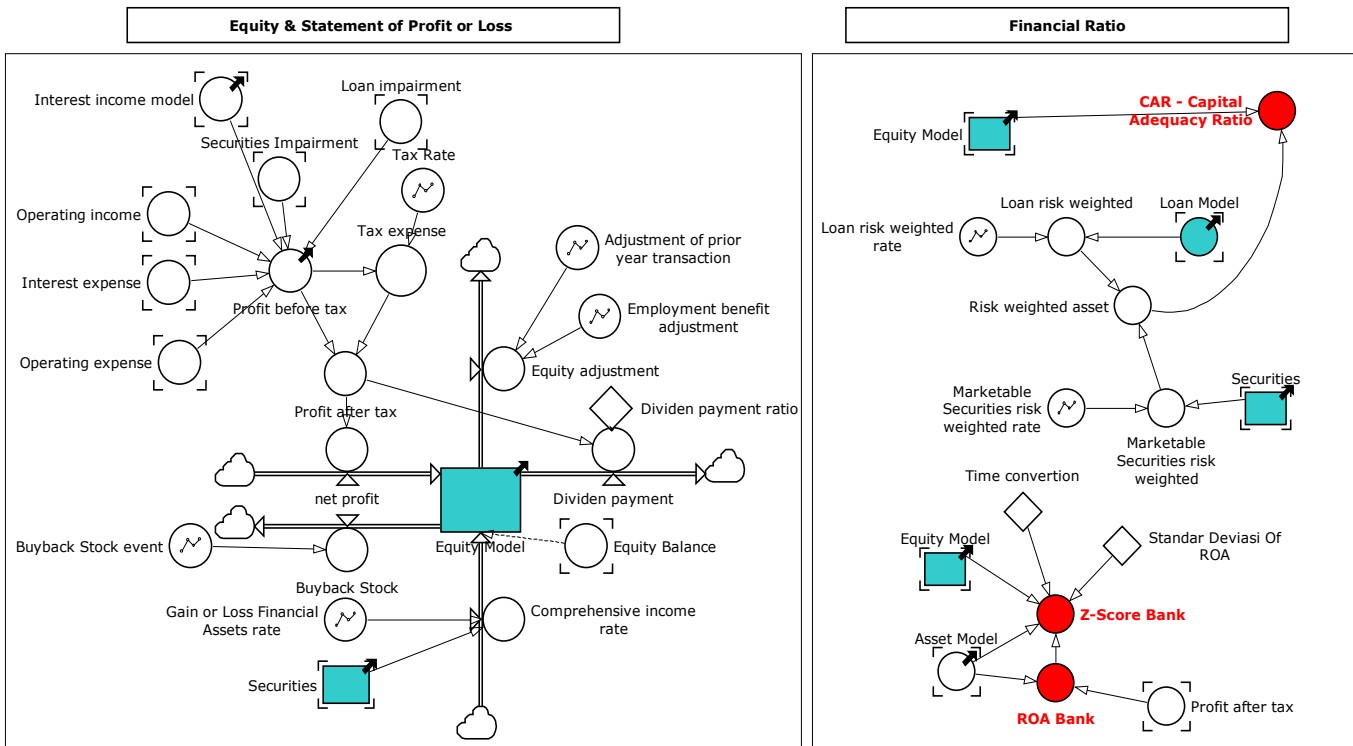

**Figure 5.** Stock-flow Diagram of Equity, Statement Profit or Loss and Financial Ratio (Source: Author).

## 4. Results and Discussion

### 4.1. Model Validation and Baseline of Bank Financial Ratios

The model was tested to determine its accuracy for further simulations. The test involved comparing the simulation results of several financial statement accounts with historical data in the 1st period (January 2020) to the 18th period (June 2021). The model output accuracy was measured using the degree of correlation (r) and the mean squared error (MSE). Table 2 shows the data from the fit model testing on the simulation results of BBRI and BMRI. The data described that the model produces financial reports with a relatively similar trend to the historical data. It has a relatively high r-value indicator of between 92.40% and 99.71% and a low mean squared error (MSE) account close to 0%. Therefore, the model is accurate for further simulation.

**Table 2.** Model Fits Testing (Source: Author).

| Account | BBRI | | BMRI | |
|---|---|---|---|---|
| | r | MSE | r | MSE |
| Loan | 99.71% | 0.004% | 98.95% | 0.009% |
| Third Party Fund | 92.40% | 0.034% | 99.68% | 0.006% |
| Equity | 99.34% | 0.010% | 99.50% | 0.013% |

The baseline financial ratio data generated by the model are referred to the projection of a bank's financial report for the period from 2020 to 2023, produced by the financial report stock-flow diagram (Figures 3–5). It is compiled based on the assumptions of loan and savings growth, interest income rate, and the interest expense rate for BBRI and BMRI. The assumptions for BBRI are as follows: third-party fund growth is 7.50%, loan growth is 7.40% per year, the average interest income rate is 9.97% per year, and interest expense rate is 2.30%. BBRI is the oldest and largest bank in Indonesia and operates 10,396 service offices with 125,602 employees. The portion of SMEs loans to total lending until the end of June 2021 reached 80.62%. BBRI is listed as the bank with the largest micro-customer base in the

world and lending to the micro-segment continues to increase because it is considered to provide higher yields.

Meanwhile, BMRI input data consist of the average third-party fund growth of 6.50%, the average loan growth of 6.45%, the interest income rate of 7.09% and the interest expense rate of 1.83% pa. BMRI's wholesale credit segment accounts for 51% of the total loans and is the driving force behind the credit growth. As of 30 June 2021 (period 18th), BMRI's office network comprises 2426 branches spread throughout Indonesia, with 38,247 employees. BMRI is the second largest bank in Indonesia in terms of total assets. The majority shareholder of BMRI and BBRI is the Government of the Republic of Indonesia.

The baseline data of BBRI and BMRI in Table 3 show that the banking solvency conditions, including the CAR and Z-Score, decreased during the COVID-19 pandemic in 2020 (period 1–12), but then increased, starting at the 18th period. The NPL ratio is still below the maximum NPL limit of 5% because it was supported by credit restructuring policies. After the policy is revoked, there is potential for NPL to arise from the high ratio of loans that are restructured to total loans, which reached 19% for BBRI and 12% for BMRI. To cover NPL risk, the bank strengthened its LLP, increasing the LLP to NPL ratio by 4.6 times in the 48th period for BBRI and 3.13 times for BMRI. The increase in reserves is to anticipate the possibility of debtors remaining in default even though credit restructuring has been carried out. To strengthen capital, it is assumed that dividends will not be distributed in the years 2022 and 2023 (from the 25th to the 48th period).

**Table 3.** Baseline Financial Ratio of BBRI and BMRI (Source: Annual Report).

| Period | BBRI | | | | | BMRI | | | | |
|---|---|---|---|---|---|---|---|---|---|---|
| | CAR | ROA | NPL | LLP | Z-Scored | CAR | ROA | NPL | LLP | Z-Scored |
| 1 | 24.02% | 0.18% | 2.06% | 2.12 | 580 | 26.41% | 0.16% | 2.39% | 1.47 | 305 |
| 3 | 20.16% | 0.15% | 2.15% | 2.74 | 518 | 22.27% | 0.25% | 2.66% | 2.34 | 254 |
| 6 | 21.51% | 0.15% | 2.20% | 2.45 | 534 | 24.79% | 0.03% | 2.52% | 2.56 | 266 |
| 9 | 21.24% | 0.13% | 2.43% | 2.42 | 536 | 23.17% | 0.09% | 2.71% | 2.54 | 261 |
| 12 | 21.65% | 0.08% | 2.70% | 2.47 | 535 | 23.44% | 0.06% | 2.92% | 2.65 | 258 |
| 15 | 23.20% | 0.12% | 3.05% | 2.53 | 570 | 23.85% | 0.19% | 3.08% | 2.54 | 265 |
| 18 | 23.15% | 0.12% | 3.23% | 2.69 | 581 | 24.19% | 0.18% | 3.14% | 2.55 | 267 |
| 21 | 23.56% | 0.12% | 3.26% | 2.97 | 592 | 24.49% | 0.17% | 3.13% | 2.60 | 270 |
| 24 | 23.22% | 0.09% | 3.23% | 3.35 | 586 | 24.84% | 0.16% | 3.10% | 2.64 | 272 |
| 27 | 23.41% | 0.16% | 3.18% | 3.56 | 589 | 25.07% | 0.13% | 3.10% | 2.70 | 274 |
| 30 | 23.77% | 0.17% | 3.16% | 3.71 | 602 | 25.32% | 0.13% | 3.09% | 2.77 | 275 |
| 33 | 24.54% | 0.19% | 3.15% | 3.87 | 626 | 25.49% | 0.14% | 3.07% | 2.85 | 276 |
| 36 | 24.75% | 0.20% | 3.14% | 4.01 | 634 | 25.65% | 0.14% | 3.06% | 2.91 | 277 |
| 39 | 25.55% | 0.20% | 3.12% | 4.17 | 659 | 25.75% | 0.15% | 3.05% | 2.97 | 280 |
| 42 | 25.88% | 0.20% | 3.10% | 4.33 | 669 | 25.82% | 0.14% | 3.04% | 3.03 | 281 |
| 45 | 26.26% | 0.20% | 3.10% | 4.47 | 679 | 26.00% | 0.15% | 3.03% | 3.09 | 283 |
| 48 | 26.55% | 0.21% | 3.09% | 4.60 | 691 | 26.08% | 0.15% | 3.03% | 3.13 | 284 |

*4.2. Early Warning on the Impact of Loan Restructuring Policy Revocation in March 2023*

In this section, the impact of the OJK Loan Restructuring Policy revocation in March 2023 is simulated on the additional potential default for outstanding Loan Restructured that gradually increase the NPL starting period (the 40th). The three scenarios for the default rate of restructuring loan balances are 10%, 20%, and 30%, which describe the bank's ability to maintain the quality of restructured loans and their effects on the banking solvency risk indicators, including NPL, ROA, CAR, and Z-Score.

As an early warning for BBRI, Figure 6 displays the results of the simulation for banking solvency risk. The simulation was carried out for the default potential on the projection of restructured loan balance at the 39th period (March 2023), which reached IDR 185 trillion. The default rate scenario consists of 30%, 20% and 10%, which describes the BBRI's ability to maintain the quality of the restructured loans. If 70% of the restructured loans can be optimally managed by worsening the collectability, then the default rate scenario is 30%, etc.

Based on the default rate scenarios of the restructured loan of 30%, 20%, and 10%, the NPL ratio will gradually increase from the 40th period until the 48th period that can reach 6.34%, 5.38% and 4.4%, or by 105.24%, 74.19% and 43% from the baseline. According to regulation, NPL should not exceed 5%. The increase in NPL has a negative impact on ROA, CAR and Z-Score, but for BBRI, this is not significant. In the 30% default rate scenario, CAR and Z-Score only decreased by 1.33% and 0.74%, respectively. This is because BBRI has an LLP ratio 4.64 times above the NPL at the baseline position. LLP absorbs the increase in impairment losses due to an increase in NPL (Figure 7). The LLP ratio decreased from 4.65 baseline position to 1.93 in the 48th period at the 30% default rate scenario. However, this LLP ratio is sufficient to cover the need for NPL elimination because it is still above the minimum ratio of 1.00 required by the regulator.

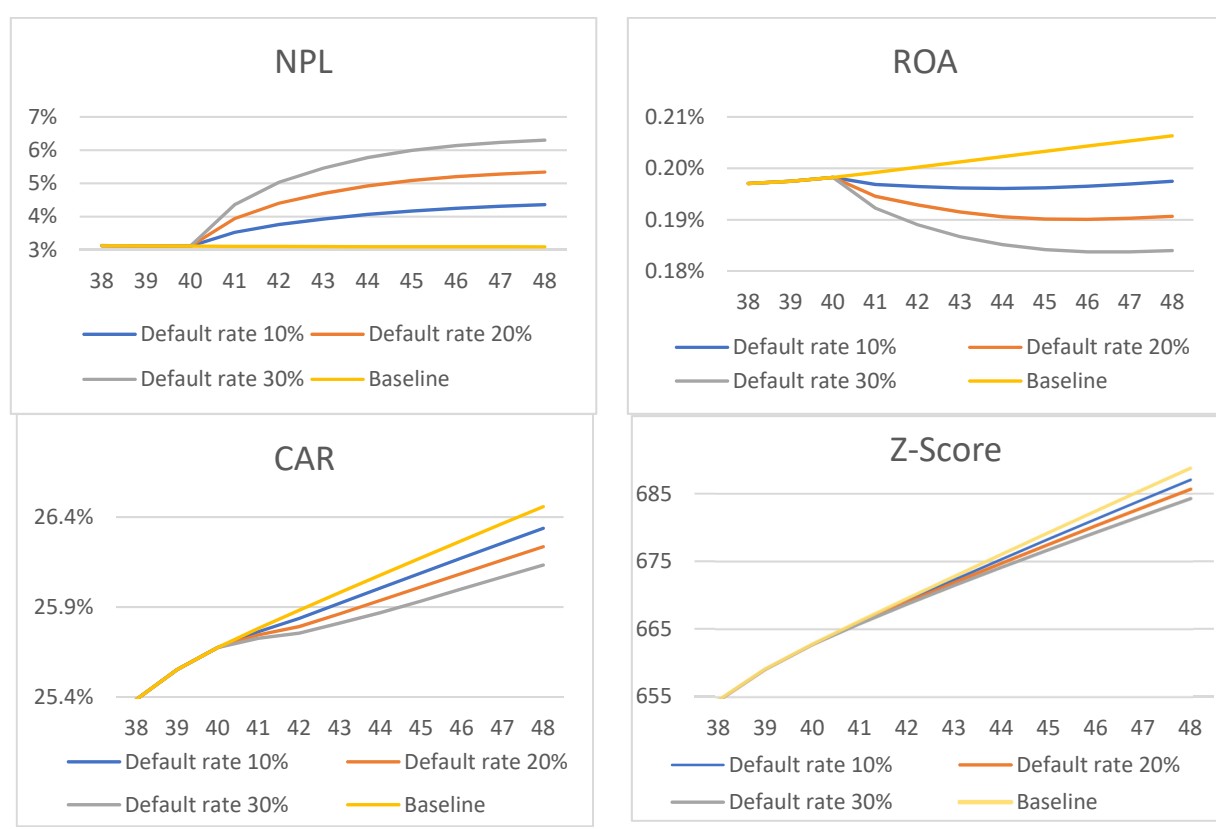

**Figure 6.** Early Warning for BBRI Solvency Risk (Source: Author).

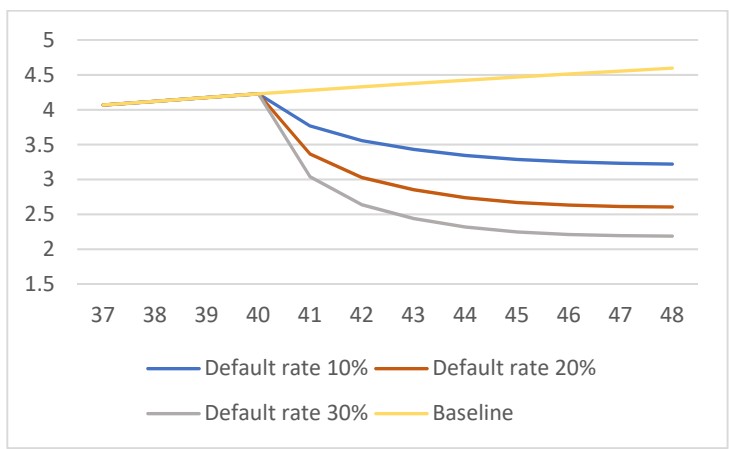

**Figure 7.** LLP of BBRI.

The solvency risk simulation of BMRI presented in Figure 8 is based on a model simulation of the loan restructured position at the period 39th (March 2023): IDR 120 trillion. The NPL ratio gradually increases up to the simulation period, namely, the 48th period (December 2023), which reached 6.25%, 5.24%, and 4.20% for the default scenario by 30%, 20%, and 10%. This scenario describes the bank's ability to control restructuring credit payments. The NPL ratio increased by 106.11%, 72.72% and 38.63% from the baseline position.

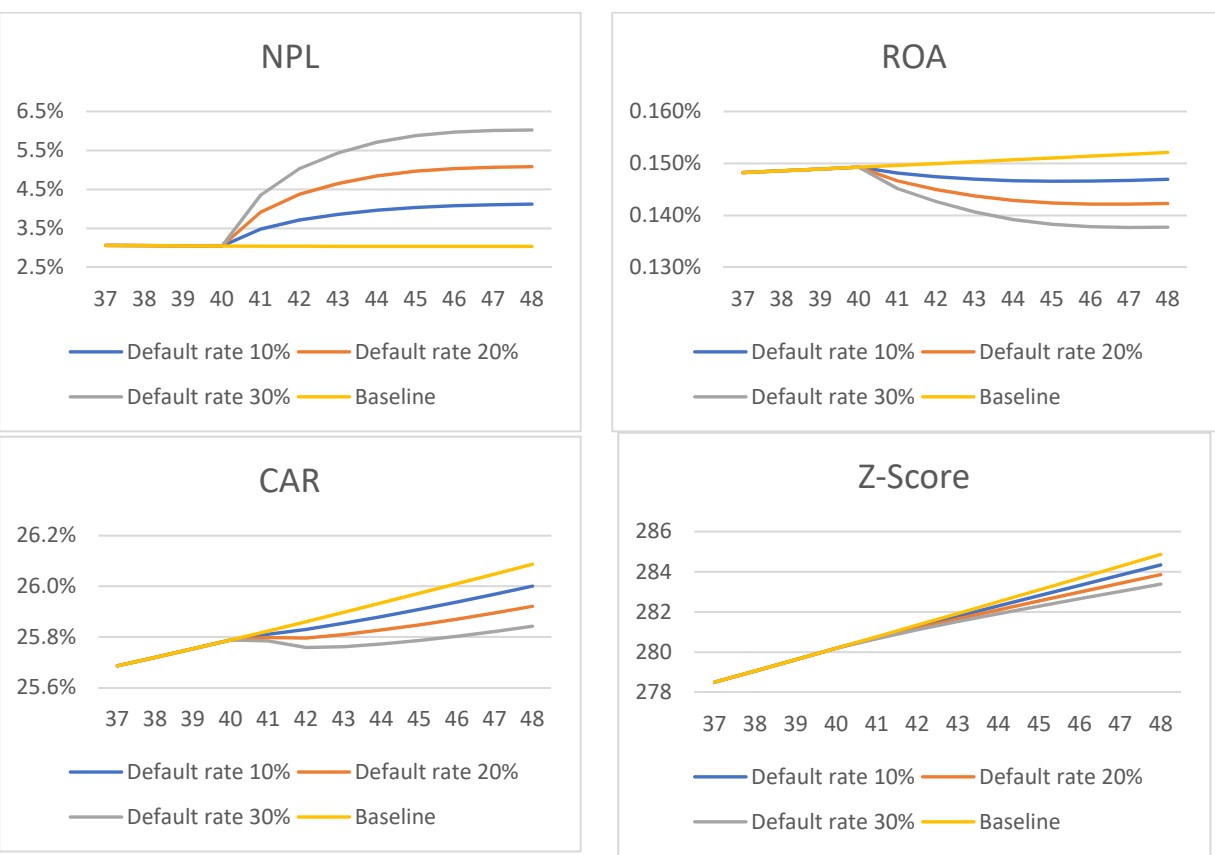

**Figure 8.** Early Warning for Solvency Risk of BMRI.

An increase in NPL will reduce profits, so that it has a negative impact on ROA, CAR and Z-Score. Based on a 30% default rate scenario, the CAR and Z-Score of BMRI ratios decreased by 1.14% and 0.70% from the baseline position, which was lower than

the increase in NPL ratio at the 48th period. Even though the CAR decreases, the ratio is still above the BMRI minimum capital requirement of 9.75%. BMRI strengthened the LLP to absorb loan impairment expense when NPL increases so that it does not harm CAR and Z-Score, as shown in Figure 9. However, the strategy of strengthening the LLP has a negative impact on the achievement of bank profits because it creates a high loan impairment expense.

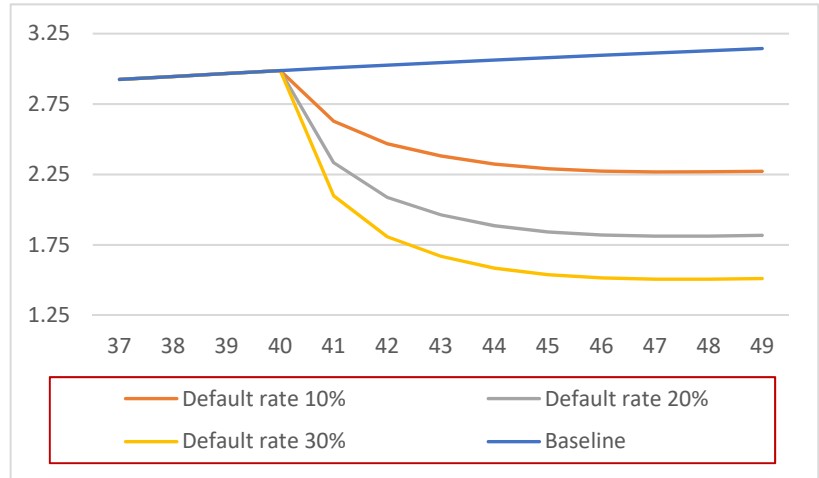

**Figure 9.** LLP of BMRI.

Based on the simulation, the solvency risk of BBRI and BMRI increased after the restructuring policy revocation, but both are still at safe solvency levels, in accordance with their respective CAR threshold. However, banks still need to strengthen the solvency level to deal with the potential default rate of a loan that is restructured by more than 30% when the credit restructuring policy is lifted in March 2023 and the economy is still recovering after the pandemic outbreak.

*4.3. Early Action to Strengthen Bank Solvency*

After the early warning simulation modeling, early action simulation is identified to strengthen the banking solvency conditions. This research led to the development of four policy options, including interest rate management, new loan policy, decreased operating expenses, and a combined policy scenario. Using certain values for the main indicators needed in each policy option, further simulations are conducted to examine the changes in potential solvency risk through the CAR and Z-Score. The changes in these parameters show that the best policy produces the highest CAR and Z-Score.

This research uses the parameters of interest expense rate and interest income rate for the interest rate management policy, loan to deposit ratio (LDR), for a new loan policy, operating expense rate, which is calculated by the percentage of total operating costs to the total loan to decrease the operating expense policy, as well as a combined policy that is a combination of the three scenarios. There is a potential for increasing NPL and decreasing CAR when the OJK loan restructuring policy ends. Based on the simulation, several policies and early actions were identified to strengthen banks' solvency. These should be implemented because the potential default rate of loan restructures could exceed 30%. Several alternative policy scenarios and examples, such as simulation results for BBRI and BMRI, are shown in Table 4.

**Table 4.** Early Action to Strengthen Bank Solvency.

| No | Policy Scenario | Policy Options (Early Action) to Strengthen Solvency for BBRI | Policy Options (Early Action) to Strengthen Solvency for BMRI |
|---|---|---|---|
| 1 | Interest rate management (policy for managing interest rates on loans and savings/deposits) | Decrease in interest expense rate (for third party funds) from around 2.66% at baseline to 2.24% per year | Decrease in interest expense rate (for third party funds) from around 1.83% at baseline to 1.63% per year |
| | | Increase in interest income rate (for new loans) from around 9.97% at baseline per year to approximately 11.24% per year | Increase in interest income rate (for new loans) from around 7.09% at baseline per year to approximately 7.97% per year |
| 2 | Increase new loan (policy to increase new loan) | Increased loan to deposit ratio from 88% at baseline to 91% | Increased loan to deposit ratio rate from 83% at baseline to 90% |
| 3 | Decreased operating expenses (policies to save bank operational costs) | Decrease in the ratio of operating expenses to total loans from 2.56% at baseline to 2.39% per year | Decrease in the ratio of operating expenses to total loans from 2.80% at baseline to 2.32% per year |
| 4 | Combined policy (policy combination) | Combination of Policy 1 to 3 | Combination of Policy 1 to 3 |

The target levels of the investigated indicators, namely loan interest rates, deposit interest rates, loan to deposit ratio (LDR) and cost to income ratios (CIR) are determined based on actual data achieved by banks at the end of 2019 or bank performance conditions prior to COVID-19. The bank's solvency level parameter data will certainly move in line with the changes of loan performance. The choice of policy scenarios is based on the simulation results for the CAR level during the simulation period, and the best policy will be selected based on the ability of the policy to generate the highest CAR ratio.

Figures 10 and 11 show the simulation results of the four early action policy scenarios for BBRI and BMRI to anticipate the increase in solvency risk when the credit restructuring policy is revoked in the 39th period. As an early action, some of the policies in Table 3 can be implemented during the period before the OJK policy is revoked, or from July 2021 (19th period) to December 2023 (48th period). An effective policy is a policy that can encourage a higher CAR and Z-Score compared to other policy options.

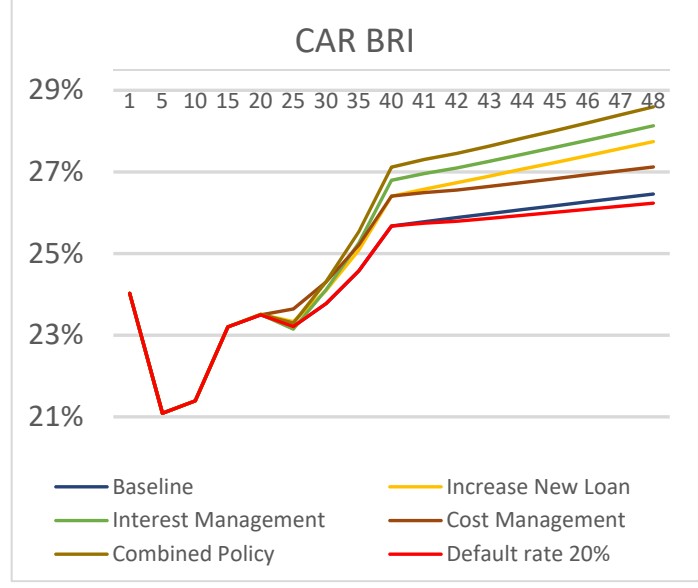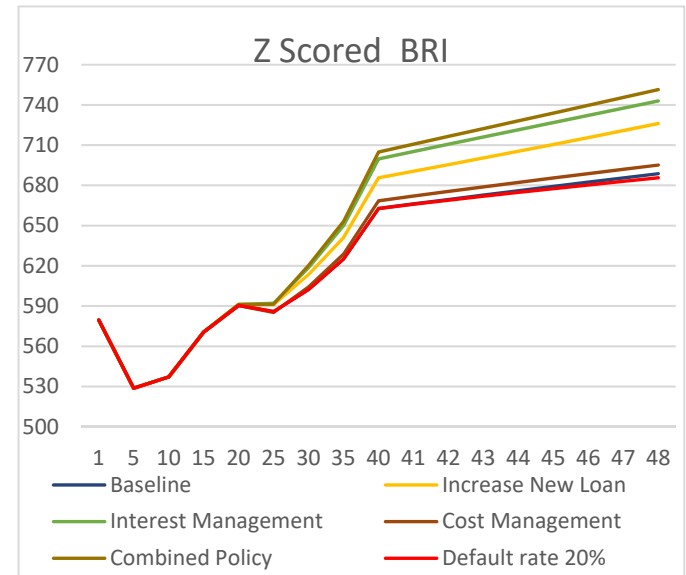

**Figure 10.** Early action Simulation of BBRI.

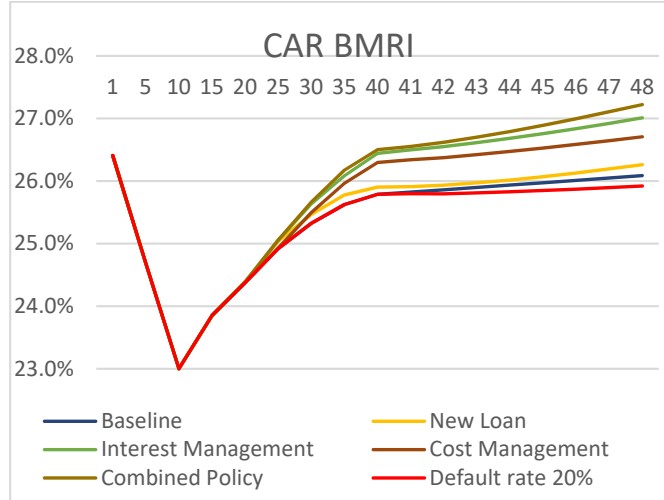
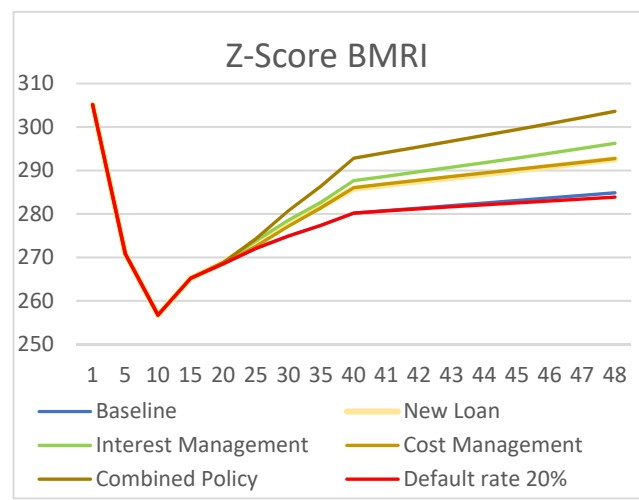

**Figure 11.** Early Action Simulation of BMRI.

At BBRI, the policy simulation of interest rate management, new loan policy, the policy of a decrease in operating expense, and combined policy scenarios resulted in higher CAR and Z-Score levels than the position at default rate 20% (Figure 10). Similarly, the Z-Score for the four policy scenario options at BBRI has the same pattern, with changes in CAR compared to the baseline level.

The combined policy pushed the CAR and Z-Score ratio of BBRI from 26.46% and 689 to 28.78% and 757, or increased by 8.76% and 9.9%, respectively, above the baseline position at the 48th month. The policies that resulted in the highest CAR and Z-Scores, after the combined policy, are the policy of decreasing operating expense, interest management policy, and new loan policy. The combined policy of increased revenue and saved costs is the best option to effectively boost CAR and Z-Score in BBRI under the recovering economic conditions caused by the COVID-19 pandemic.

For BMRI, based on the two parameters of CAR and Z-Score on Figure 11, the combined policy causes the highest changes in CAR and Z-Score values, followed by the policy of decreasing operating expenses, interest management policy, and new loan policy. Moreover, the combined policies strengthen the solvency of BMRI compared to other policies. Therefore, the combination of policies to increase revenue and save costs is the best option to effectively boost CAR in BMRI under abnormal economic conditions, due to the impact of the pandemic outbreak.

The combined policy also has a high sensitivity level to boost BMRI's CAR and Z-Score by 20.23% and 24.56% from the baseline position for the 48th period. In contrast, policies with low leverage in the current economic conditions are new loan policies. This is because the demand for new loans is still low, due to the COVID-19 pandemic.

*4.4. Discussion*

Several important points should be discussed regarding the simulation of the loan restructuring policy revocation that began in March 2020. The first is to evaluate the potential of the default on the restructured loans. This evaluation allows banks to control the collectability of restructured loans to avoid them deteriorating or becoming NPLs (Bauer et al. 2021). However, evaluation should be monitored through the bank guidance to the debtors with the potentially non-performing loans. This could be implemented by visiting the debtors and their businesses, or conducting some on-site monitoring, to analyze the respected debtors' ability to pay. There should be some concerns from banks regarding the provision of advice to debtors regarding business management to maintain and improve the payment capacity of the restructured loans. Therefore, loan monitoring by

banks suppresses non-performing loans and improves financial performance (Duong et al. 2020; Hidayat et al. 2021).

In a normal situation (not a pandemic), the increase in loan interest rates and a decrease in deposit rates will greatly depend on the level of price elasticity of each product. In this case, if the loan interest rate decreases, the demand for loan from the prospective debtors will increase, and if the deposit interest rate decreases, deposit placement activities will decrease.

The COVID-19 condition has led to government intervention in handling the situation through restrictions on community activities. This is the main reason that causes less than optimal new loan growth or in other words, although loan interest rates decreases, the demand of prospective debtors obtaining the new loan does not increase. This is because people still in doubt of their ability to repay loans.

However, the design of this study has considered the factors that influence the rate of loan growth (additional loan) with the following explanation:

- The increase in new loans is not only influenced by loan interest rates, which tend to decline during the pandemic, but is greatly influenced by the COVID-19 condition with the level of public and business trust as potential debtors being quite low due to the tightening economic activities and doubts about their ability to loan repayment.
- The increase in new loans is influenced by the level of bank liquidity, which was quite abundant during a pandemic. However, with the level of trust from the public and business that had not recovered as well as the economic activity that had not yet recovered, the bank could not carry out the new loan growth optimally.

Taking into account the simulation results, it can be conveyed that in the COVID-19 period there are conditions, namely: abundant bank liquidity, declining loan interest rates and declining deposit interest rates. This can then be adjusted to post-COVID-19 conditions, where the loan interest rate can be increased, namely from a declining interest rate to the original interest rate, in line with increasing public and business confidence in the conditions of post COVID-19 and the recovery in economic activity.

The simulation results also show that the banks have a fairly good level of LLP and CAR to absorb the solvency risk. Therefore, although there is an increase in NPL, this does not have a significant impact on CAR and Z-Score (Agenor and Zilberman 2015). A high level of LLP reduces the bank solvency risk but increases the loan impairment losses, and suppresses profit and capital growth. Therefore, banks face a trade-off between solvency and profitability (Zheng et al. 2019).

There is a challenge in the simulation of policy scenarios by banks to prepare better action plans in response to worsening NPLs when the restructuring policy is revoked. This allows for the impact of the policy on the bank solvency to be known before it is implemented (Paiva et al. 2020; Asadollahi et al. 2021). Moreover, it enables banks to prepare for the existing policy scenario options (Morecroft 2015).

The model has been developed into a self-simulation facility that could be implemented by the bank management to enable banks to deal with the potential solvency risks using their own data. Banks could calculate the level of solvency and other risks as a consequence of the COVID-19 pandemic through a simple interface that generates early warning and early action. Figure 12 shows the Bank Solvency Risk Self-Simulation Interface developed in this research. Three parts were prepared in the application. The first part shows data input instruments for potential defaults on restructured loans. The second part shows input instruments for bank policy parameters to manage solvency risk through four policy scenarios. The third part displays the simulation results of bank solvency indicators, including NPL Ratio, CAR, ROA and Z-Score.

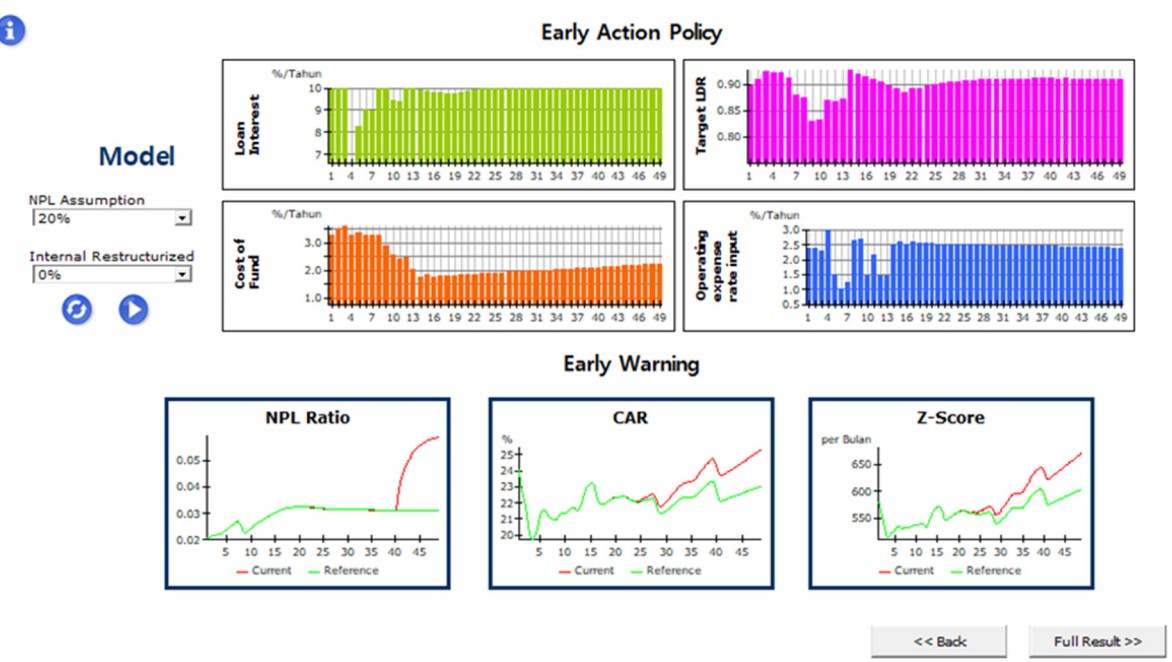

**Figure 12.** Bank Solvency Risk Self-Simulation Interface Developed in the Research.

The implementation of a self-simulation facility requires bank officers to provide input data or assumptions regarding the estimated percentage of restructured loan defaults and the potential non-performing loans. These inputs produce an early warning about solvency risk. Furthermore, bank management could prepare inputs for each solvency risk management policy provided by applying the required parameters. After obtaining the input data for the four policies, the application generates new conditions, providing the policy by displaying the NPL, CAR, and Z-Score as early actions for the bank management.

Based on the early warning and early action information, using self-simulation, the bank could implement the policy that produces the highest CAR and Z-Score at the first opportunity. The earlier the best policy is implemented, the better prepared banks become to face the loan restructuring policy revocation that will take place in March 2023. After the policy is implemented, and during the lead up to the policy revocation, banks could still perform self-simulation. However, they should consider the current conditions of loan performance through the NPL ratio. They could also continuously monitor or update bank's readiness to face solvency risk after the revocation of the OJK loan restructuring policy.

## 5. Conclusions

This article develops an early warning system model for bank bankruptcy risk and a simulation of management policy response. The development of the model with a system dynamics methodology adds theoretical knowledge about the flow of the bank financial transactions that caused the bank bankruptcy risk, and then produces simulations and management policy choices to reduce the potential risks. By identifying the flow in the system dynamics, bank management can develop appropriate risk mitigation policy choices according to the symptoms that have arisen.

System dynamics methodology is used to build models and perform early warning and early action simulations to help banks identify solvency risks earlier and design policies and strategies to mitigate the risk. Early warnings regarding bank solvency risk could be identified through simulations and early actions could be taken to overcome the potential risks and problems after the restructuring policy revocation. To deliver the robust model of solvency risk, this research used the publicly accessed data from two banks and analyzed some financial ratios based on the financial statements. To enable banks to use their data,

this research provided a self-simulation facility, so that bank management could obtain a more accurate solvency risk prediction. It is suggested that future research could consider not only solvency risk but also the liquidity and market risks.

The limitations of the analysis that carried out in this paper, among others, that it only discussed the potential loan risk as the beginning of the bankruptcy risk and has not discussed the other causes of the loan risks such as the debtor moral hazard, the weak analysis of creditworthiness, the declining of the payment capacity of the debtors due to the pandemic COVID-19 and the specific economic sectors that experienced the high level of loan risks. It is suggested that future research could consider not only the loan and solvency risk but also the liquidity and market risks in the model. Further, the upcoming research is suggested to discuss the causes of the bankruptcy risks that have not been analyzed in this article.

**Author Contributions:** T.H. carried out the conception, design of the work, and drafting the article. D.M. reviewed the literature, methodology and gave final approval of the article. S.R.N. reviewed data analysis and interpretation. F.A. worked on data collection and the modelling. M.A.N.S. worked on the simulation. All authors have read and agreed to the published version of the manuscript.

**Funding:** The research presented in this scientific article was self-funded.

**Data Availability Statement:** The data presented in this study are available on request from the corresponding author.

**Conflicts of Interest:** The authors declare no conflict of interest.

## Appendix A

**Table A1.** Formula of Variable in Stock-flow Diagram.

| Variabel | Formula |
|---|---|
| Performing Loan | Initial Performing Loan + (Loan Payment − Additional Loan) + (NPL Restructuring − Additional NPL) − Performing Loan to Restructured Loan |
| Loan Payment | (Loan Maturity ∗ Performing Loan) ∗ (1 + Seasonal maturity) |
| Additional Loan | (MIN (Max Liquid Asset Outflow, If ((Marketable Securities Model + Liquid Asset Model)>Expected liquid Asset, Performing loan correction, 0<<IDR Million/Month>>) ∗ (Credit Impact in COVID situation/Ratio interest rate loan to its delayed effect))) |
| Additional NPL | Performing loan ∗ Realized NPL Rate Average by Sectors |
| NPL Restructuring | (MAX (Target NPL Restructuring, (Target NPL Restructuring + NPL Correction))/Time to Restructuring) ∗ Loan Model |
| NPL | Initial NPL + (Additional NPL − NPL Restructuring) + Rest Loan to NPL − NPL Write off |
| Rest Loan to NPL | MIN (Maximum Restructurized Loan Outflow, decrease on total loan cumulative) |
| NPL write off | (Non performing loan NPL ∗ NPL write-off rate) |
| Restructurized Loan | Initial Restructurized Loan + Performing Loan to Restructured Loan − Rest Loan to NPL − Restructurized Loan Payment |
| Restructurized Loan Payment | MAX(0<<IDR Million/month>>,MIN(Maximum Restructurized Loan Outflow, Loan Maturity/Multiplier Maturity Rate in Restructurized Implemented ∗ Restructurized Loan)) |
| Liquid Asset Model | Initial Liquid Asset + Loan Payment − Additional Loan + Restructurized Loan Payment + Cash Inflow − Cash Outflow − Buy Marketable Securities (MS) − Sell MS |
| Cash Inflow | Interest income + Operating income + New borrowing + Additional TPF + NPL Write Off Paid + Fixed Asset Disposals |

**Table A1.** *Cont.*

| Variabel | Formula |
| --- | --- |
| Cash Inflow | (Operating expense − Depreciation)+ Additions of Fixed Asset + Borrowing Payment + Withdrawal TPF + Interest Expense + Dividen Payment + Tax expense + Buyback Stock |
| Sell MS | MIN (Maximum Sell of Marketable Securities, Indicated to sell MS ∗ LDR to MS Ratio Sell) |
| Buy MS | MIN ((Indicated to buy MS ∗ LDR to MS Ratio Buy)∗1+Irreguler policy of investment, Max Liquid Asset ouflow) |
| Securities | Initial Securities + Buy MS − Sell MS + Gain of Value |
| Reserve of Loan Impairment Model | Initial of Reserve of Loan Impairment Model + Loan Impairment − Impairment Outflow + Adjustment of Loan Impairment |
| Loan Impairment | (Loan Model ∗ Impairment Rate) |
| Impairment Outflow | (MIN (Maximum Loan Impairment Available, NPL write off)) |
| NPL Ratio | Non performing loan NPL/Loan Model |
| Loan Loss Provision (LLP) | Reserve of Loan Impairment Model/Non performing loan NPL |
| Third Party Fund (TPF) | Initial TPF + Additional TPF − Withdrawal TPF |
| Additional TPF | TPF national growth rate ∗ Seasonal TPF ∗ (Market Share Normal TPF ∗ Effect of market share from asset) |
| Withdrawal TPF | (Third party fund TPF Model/Time of TPF Withdrawal) ∗ (Seasonal withdrawal) |
| Equity Model | Initial Equity + Net Profit − Equity Adjustment − Dividend Payment − Buy back of Stock |
| Net Profit | (Interest Income − Interest Expense + Operating Income − Operating Expense − Loan Impairment -Tax expense) |
| Equity Adjustment | Adjustment of prior year transaction + Employment Benefit Adjustment |
| Dividend Payment | Profit After Tax ∗ Dividend Payout Ratio |
| Buy Back of Stock | Buy back decision or event |
| Capital Adequacy Ratio (CAR) | Equity Model/Risk Weighted Asset ∗ 100<<%>> |
| ROA | Profit After Tax/Asset Model |
| Z-Score | (ROA + (Equity Model/Asset Model))/Standard Deviation of ROA |

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
