# Peer review of "Early Warning Early Action for the Banking Solvency Risk in the COVID-19 Pandemic Era: A Case Study of Indonesia"

_economies, doi:10.3390/economies10010006_

Round 1

Reviewer 1 Report

My comments on the  paper - Early Warning Early Action for the Banking Solvency Risk Responses to the Loan Restructuring Policy Extension in the COVID-19 Pandemic Era: A Case Study of Indonesia - are as follows.

 The paper presents an interesting analysis and we consider that the research is of interest. However, as it stands, the paper needs revisions.

The title of the research is quite vague and too long.

The abstract is clear, presents the purpose of the paper. The keywords are appropriately chosen.

The introduction provides the necessary background information and states the objectives.

In the introduction section, the structure of the paper on sections is not provided.

The author does not mention the added value of the paper. In the introduction it must be stated the added value that the paper brings to the existing academic literature.

The analysis of the literature is limited.

The structure of the paper is appropriate.

The research methodology used by the author is adequate for the approached subject.

We recommend indicating the references for the tables and figures.

We recommend the development of the Conclusions section. In addition, We consider that the author can show the limitations of the analysis carried out in his paper.

Author Response

The paper presents an interesting analysis and we consider that the research is of interest. However, as it stands, the paper needs revisions.

  1. The title of the research is quite vague and too long.

Response:

The title of the research is corrected to be “Early Warning Early Action for the Banking Solvency Risk in the COVID-19 Pandemic Era: A Case Study of Indonesia”.

  1. The abstract is clear, presents the purpose of the paper. The keywords are appropriately chosen.
  2. The introduction provides the necessary background information and states the objectives.
  3. In the introduction section, the structure of the paper on sections is not provided.

Response:

The structure of this paper consists of 5 parts, beginning with the introduction that contains the background of the research. Section 2 discusses the literature review, namely the related literature and previous research, section 3 explains the research methodology followed by the research results and section 4 is a discussion of the results of the research. Lastly is the section of the conclusion and suggestions that also contains the implication of the research results. This study is ended with a reference of the studies that used in the research.

  1. The author does not mention the added value of the paper. In the introduction it must be stated the added value that the paper brings to the existing academic literature.

Response:

This research makes an important contribution to the study of the early warning system of bank bankruptcy risks for several reasons, namely: (1) the use of system dynamics simulation methods to modelling the complex, dynamic and ongoing bank risk behaviour during the pandemic Covid-19, (2) the research is to be able to produce an effective early warning of the bank solvency risks and (3) the research is a kind of a forward looking oriented simulation to predict the potential bank bankruptcy risks. The added value of this research is the existence of a dynamic bank balance sheet simulation so that the condition of the bank's asset, liabilities, capital and profit and loss during the Covid-19 pandemic and the risk of bank solvency can be detected at any time. By monitoring the condition of bank’s loan performance through several main variables related to solvency risk, bank’s management can determine appropriate policies to reduce this risk and indirectly reduce the risk of bank’s bankruptcy. This research is able to produce simulation models in the form of early warnings and simulations of several policy options as early actions for the bank’s management to response the risk of non-performing loan, solvency risk and bank’s bankruptcy due to the Covid-19 pandemic.

  1. The analysis of the literature is limited.

Response: the improved literature review is as follows.

According to Vany, A. S. de. (1984), the main causes of bank bankruptcy are the information asymmetry, agency problem and moral hazard that occur together. Smith, AD (2010) found evidence that there is a correlation between agency problems and bank bankruptcy in the crisis period of 2007 & 2008. The agency problem is the problem of mismatch of interests between shareholders as principals and management as agents (Jensen and Meckling, 1976; Rose, P. , 1992). In the banking sector, agency relationships occur between bank management with shareholders and banking supervisory authorities (Henrard, L., & Olieslagers, R. 2004) as well as with depositors (Kuritzkes, A., Schuermann, T. and Weiner, SM 2003). Banking supervisory authorities play a role in protecting the interests of depositors by issuing various regulations that must be obeyed by bank management and shareholders (Donnellan, JT and Rutledge, W., 2016), including the issuance of policies for loan restructuring during the COVID-19 pandemic by the Financial Services Authority, then it is an agency relationship intervention in order to reduce the risk of bank bankruptcy and protect depositors (Hidayat et.al, 2021).

Bank bankruptcy can also arise due to changes in financial conditions both internally and externally of the bank. Bank bankruptcy can also arise from increased loan risk arising from debtor moral hazard, weak analysis of creditworthiness, external conditions such as the decline in the community's economic capacity due to the COVID-19 pandemic or lending to high-risk sectors. Bank management as an agent in the agency theory needs to recognize weak signals in the economic environment that will affect loan risk and bank bankruptcy, such as high NPL levels and declining CAR ratios. According to Ansoff, H. I. (1975), a weak signal is a symptom of bank performance that provides the basis for managerial decision making to ensure that the bank's strategy can be achieved. Meanwhile, based on the weak banking signal information, a system is needed to provide some early warnings for bank management to be aware of the potential risks that may arise. Early warning systems are a key tool for bank management to anticipate and make policies to reduce the potential risks of bank bankruptcy (Ginoglou, D., Agorastos, K., 2002; Curry, Timothy J. and Fissel, Gary S. and Elmer, Peter J., 2004; Gunnersen, SE (2014).

Most of the literature research on Early Warnings of bank solvency risk is backward-oriented. This means that the literature is based on historical financial statements to provide Early Warning indicators of these potential risks. For instance, Korzeb and NiedzióÅ‚ka (2020); Barua and Barua (2021); and Hardiyanti and Aziz (2021) showed the phenomenon of increased NPL risk during the COVID-19 pandemic. The increase in NPL reduced cash flow, profit, and CAR (Mayes and Stremmel 2012; Donellon and Rutledge 2016). Consequently, a decrease in CAR increased the bank solvency risk, as measured by Z-Score (Lepetit et al. 2020). Existing research is quite limited to produce predictors for bank bankruptcy risks, and has not modeled bank management policies to minimize this risk. 

Facing the COVID-19 pandemic requires a forward-looking approach for Early Warnings of potential bank solvency risks and Early Action to prevent these risks. For these reasons, the simulation methodology was used to develop baseline projections for financial statements. These baselines describe future financial risk conditions and changes in their behavior due to some unexpected events. Furthermore, the simulation results were used to develop several alternative Early Action policies to reduce solvency risk (Pavlov and Katsamakas 2021; Petropoulos et al. 2020). These include the promotion of loan growth, managing restructured loans, increasing bank operational cost efficiency, lowering interest expenses, and increasing loan interest (Bastana et al. 2016; Borys et al. 2019; Rahmi and Sumirat 2021). Those policies are expected to strengthen bank capital to avoid solvency risk. However, the Early Action policy should be simulated first to determine the feedback on potential changes in the solvency risk levels and the optimum policy options (Schuermann. 2014). According to Wu (2014) and Kunc et al. (2018), the system dynamics methodology is a simulation modeling that accommodates the feedback process in managerial decision making.

System Dynamics is a methodology to design strategies and policies with computer simulation tools (Sapiri et al. 2020), to produce better responses to the complex and dynamic problems in the social, managerial or economic fields (Sterman 2000; Morecroft 2015; Duggan 2016). To solve the complex problems, according to Bala et al. (2017), the structures, and the relationship between the structures in the problem, should be analyzed. The System Dynamics model describes the structure of financial statement accounts based on Stock, Rate, Auxiliary, and Constant. The pattern of the relationships between the accounts in the financial statements is modeled through causal-loop and stock-flow diagrams (García 2019). The financial statements model of system dynamics was developed in some studies by Islam et al. (2013); Wu (2014); Istiaq (2015); Pierson (2020); Aksu and Tursun (2021); Pavlov and Katsamakas (2021) and Hidayat et al. (2021) to analyze financial reporting, banking risk management, management control systems, and solvency stress-testing.

  1. The structure of the paper is appropriate.
  2. The research methodology used by the author is adequate for the approached subject.
  3. We recommend indicating the references for the tables and figures.

Response: the indicating of the references for the tables and figures have been added in the revised manuscript.

  1. We recommend the development of the Conclusions section. In addition, we consider that the author can show the limitations of the analysis carried out in his paper.

Response: the improved conclusions section is as follows.

This article develops an early warning system model for bank bankruptcy risk and a simulation of management policy response. The development of the model with a system dynamics methodology adds theoretical knowledge about the flow of the bank financial transactions that caused the bank bankruptcy risk, and then produces simulations and management policy choices to reduce the potential risks. By identifying the flow in the system dynamics, bank management can develop appropriate risk mitigation policy choices according to the symptoms that have arisen.

System dynamics methodology is used to build models and perform Early Warning and Early Action simulations to help banks identify solvency risks earlier and design policies and strategies to mitigate the risk. Early Warnings regarding bank solvency risk could be identified through simulations and Early Actions could be taken to overcome the potential risks and problems after the restructuring policy revocation. To deliver the robust model of solvency risk, this research used the publicly accessed data from two banks and analyzed some financial ratios based on the financial statements. To enable banks to use their data, this research provided a self-simulation facility, so that bank management could obtain a more accurate solvency risk prediction.

The limitations of the analysis that carried out in this paper, among others, that it only discussed the potential loan risk as the beginning of the bankruptcy risk and has not discussed the other causes of the loan risks such as the debtor moral hazard, the weak analysis of creditworthiness, the declining of the payment capacity of the debtors due to the pandemic COVID-19 and the specific economic sectors that experienced the high level of loan risks. It is suggested that future research could consider not only the loan and solvency risk but also the liquidity and market risks in the model. Further, the upcoming research is suggested to discuss the causes of the bankruptcy risks that have not been analyzed in this article.

Reviewer 2 Report

The paper addresses an important and topical issue related to the management of solvency risk in the banking sector. The principal advantage of the study stems from its practical applicability, which makes is potentially appealing not only for academics, but also for bank managers and representatives of supervisory institutions, as the Authors develop and apply a financial simulation model to examine the impact of various factors on bank solvency risk on the example of two largest commercial banks in Indonesia. It also worth to point out the timeliness of the research, as it attempts to explore the consequences of the expected revocation of the extraordinary policy regarding the treatment of non-performing loans adopted in 2020 in response to the COVID-19 pandemic by the Indonesian Financial Services Authority.

Notwithstanding the above, and the overall soundness of the research design, the Authors should address the following concerns:

  • in Figure 2 (p. 4), risk-weighted assets are positively linked to CAR, however given the fact that risk-weighted assets constitute the denominator of CAR, their increase, ceteris paribus, naturally leads to decrease in CAR. Moreover, according to the diagram, the level of total loans exerts a direct negative impact on CAR. Therefore, it seems that the above linkages should be rearranged to improve the overall clarity of the model. First, total loans should be positively linked to risk-weighted assets, and next, risk-weighted assets should be negatively linked to CAR. In addition, the diagram seems to ignore the impact of bank dividend policy on the examined financial relationships,
  • in table 4 (pp. 12-13) the Authors present various policy scenarios aimed at strengthening solvency of each examined bank. It is unclear however, how the target levels of the investigated indicators have been determined. Additionally, it seems that some of the suggested policies do not take into account the response of the market. For instance, if the banks were to simultaneously decrease the interest rates on deposits and increase the rates on loans, it would likely result in reduction of both their deposit bases and loan extension capabilities, and thus affect the assumptions regarding the rates of growth in deposits and loans employed in the model (see p. 8). Moreover, In the 'Combined Policy Scenario' the Authors suggest that the banks should simultaneously increase interest rates on loans, decrease interest rates on deposits, and increase the loan to deposit ratios. The possibility of achieving such results is, however, largely dependent on the price elasticities of demand for deposits and loans. Given the above the Authors should address the above concerns both in the research design and the discussion of results,
  • the formula for LLP ratio on p. 4 requires correction,
  • in Figures 6 and 7 (pp. 10-11) it would be advisable to keep the colours of the lines corresponding to the same default rate scenarios constant to improve the coherence of the analysis,
  • the manuscript seems to require some additional proof reading with respect to grammar, wording and style. In particular, it seems highly advisable to review and correct the references to financial items and ratios throughout the paper, see e.g.:
    • ‘growth of loan’ instead of ‘growth of loans’ (Figure 1, p. 3),
    • ‘solvable’ instead of ‘solvent’ (p. 4),
    • ‘Return on Asset’ instead of ‘Return on Assets’ (p. 4).
    • ‘Total risk-weighted asset’ instead of ‘Total risk-weighted assets’ (p. 4)
    • ‘Non-performing loan (NPL) ratio’ instead of ‘Non-performing loans (NPS) ratio’ (p. 4),
    • ‘Total loan’ instead of ‘Total loans’ (p. 4),
    • ‘Z-scored’ instead of ‘Z-score’ (Table 3, p. 9).

Author Response

Notwithstanding the above, and the overall soundness of the research design, the Authors should address the following concerns:

in Figure 2 (p. 4), risk-weighted assets are positively linked to CAR, however given the fact that risk-weighted assets constitute the denominator of CAR, their increase, ceteris paribus, naturally leads to decrease in CAR. Moreover, according to the diagram, the level of total loans exerts a direct negative impact on CAR. Therefore, it seems that the above linkages should be rearranged to improve the overall clarity of the model. First, total loans should be positively linked to risk-weighted assets, and next, risk-weighted assets should be negatively linked to CAR. In addition, the diagram seems to ignore the impact of bank dividend policy on the examined financial relationships,

Response:

Figure 2 (updated) as above regarding the Causal-loop Diagram of the Early Warning Early Action Model has been corrected. As suggested, the figure shows a more precise relationship between total loans, which is positively correlated with risk-weighted assets, and furthermore, risk-weighted assets are negatively correlated with CAR. However, it can be conveyed that in the calculation of the capital adequacy ratio (CAR) and dividend policy simulation, the initial manuscript has been carried out correctly with reference to Figure 5 (regarding Stock-flow Diagram of Equity, Statement of Profit or Loss and Financial Ratio) and Table A1 (capital adequacy ratio/CAR formula) and equity model formula).

in table 4 (pp. 12-13) the Authors present various policy scenarios aimed at strengthening solvency of each examined bank. It is unclear however, how the target levels of the investigated indicators have been determined.

Response:

   The target levels of the investigated indicators, namely loan interest rates, deposit interest rates, Loan to Deposit Ratio (LDR) and Cost to Income Ratios (CIR) are determined based on actual data achieved by banks at the end of 2019 or bank performance conditions prior to COVID-19. The bank's solvency level parameter data will certainly move in line with the changes of loan performance. The choice of policy scenarios is based on the simulation results for the CAR level during the simulation period, and the best policy will be selected based on the ability of the policy to generate the highest CAR ratio.

Additionally, it seems that some of the suggested policies do not take into account the response of the market. For instance, if the banks were to simultaneously decrease the interest rates on deposits and increase the rates on loans, it would likely result in reduction of both their deposit bases and loan extension capabilities, and thus affect the assumptions regarding the rates of growth in deposits and loans employed in the model (see p. 8). Moreover, In the 'Combined Policy Scenario' the Authors suggest that the banks should simultaneously increase interest rates on loans, decrease interest rates on deposits, and increase the loan to deposit ratios. The possibility of achieving such results is, however, largely dependent on the price elasticities of demand for deposits and loans. Given the above the Authors should address the above concerns both in the research design and the discussion of results,

Response in the research design:

Figure 3. Stock-flow Diagram of Financial Assets

In the research design, the impact of changes in loan interest rates on additional loans is explained in the function of the interest rate loan to its delayed effect ratio variable in the additional loan formula in Table A1. If the annual interest rate of loan is increased, it will have a negative impact on additional loans.

Furthermore, for deposit growth, we use the variable size of bank assets. This is based on the Indonesia Banking Survey 2017 conducted by PWC.  It can be seen that the amount of bank assets has a strong correlation with additional third-party funds. Furthermore, additional loans at Bank BRI and Bank Mandiri are encouraged because they have more extensive networks and access to customers. Therefore, in this study the effect of market share from assets is used as a reinforcing variable for additional third-party funds. Moreover, most of the third-party funds of Bank BRI and Bank Mandiri come from the government institutions and state-owned companies which are not sensitive to the amount of interest rates on deposit. The explanation is shown in the following figure (Figure 4. Stock-flow Diagram of Liabilities).

Figure 4. Stock-flow Diagram of Liabilities.

Response in the Discussion of Result

In a normal situation (not a pandemic), the increase in loan interest rates and a decrease in deposit rates will greatly depend on the level of price elasticity of each product. In this case, if the loan interest rate decreases, the demand for loan from the prospective debtors will increase, and if the deposit interest rate decreases, deposit placement activities will decrease.

The COVID-19 condition has led to the Government intervention in handling the situation through restrictions on community activities. This is the main reason that causes less than optimal new loan growth or in other words, although loan interest rates decreases, the demand of prospective debtors to get the new loan does not increase. This is because people still in doubt of their ability to repay loans.

However, the design of this study has considered the factors that influence the rate of loan growth (additional loan) with the following explanation:

  • The increase in new loans is not only influenced by loan interest rates, which tend to decline during the pandemic, but is greatly influenced by the COVID-19 condition with the level of public and business trust as potential debtors being quite low due to the tightening economic activities and doubts about their ability to loan repayment.
  • The increase in new loans is influenced by the level of bank liquidity, which was quite abundant during a pandemic. But with the level of trust from the public and business that had not recovered as well as the economic activity that had not yet recovered, the bank could not carry out the new loan growth optimally.

Taking into account the simulation results, it can be conveyed that in the COVID-19 period there are conditions, namely: abundant bank liquidity, declining loan interest rates and declining deposit interest rates. This can then be adjusted to post-COVID-19 conditions, where the loan interest rate can be increased, namely from a declining interest rate to the original interest rate, in line with increasing public and business confidence in the conditions of post COVID-19 and the recovery in economic activity.

the formula for LLP ratio on p. 4 requires correction,

Response: the formula for LLP ratio on p.4 has been corrected in the revised manuscript.

in Figures 6 and 7 (pp. 10-11) it would be advisable to keep the colours of the lines corresponding to the same default rate scenarios constant to improve the coherence of the analysis,

Response: the advise to keep the colours of the lines corresponding to eht same default rate scenarios constat to improve the coherence of the analysis is done in the revised manuscript.

the manuscript seems to require some additional proof reading with respect to grammar, wording and style. In particular, it seems highly advisable to review and correct the references to financial items and ratios throughout the paper, see e.g.:

‘growth of loan’ instead of ‘growth of loans’ (Figure 1, p. 3),

‘solvable’ instead of ‘solvent’ (p. 4),

‘Return on Asset’ instead of ‘Return on Assets’ (p. 4).

‘Total risk-weighted asset’ instead of ‘Total risk-weighted assets’ (p. 4)

‘Non-performing loan (NPL) ratio’ instead of ‘Non-performing loans (NPS) ratio’ (p. 4),

‘Total loan’ instead of ‘Total loans’ (p. 4),

‘Z-scored’ instead of ‘Z-score’ (Table 3, p. 9).

Response: the initial manuscript has been That the initial manuscript has used proof reading services provided by MDPI (certificate attached).

Submission Date

23 November 2021

Date of this review

10 Dec 2021 13:00:44

Submission Date of the improved manuscript: 18 Dec 2021
